# Enzymatically Digested Food Waste Altered Fecal Microbiota But Not Meat Quality and Carcass Characteristics of Growing-Finishing Pigs

**DOI:** 10.3390/ani9110970

**Published:** 2019-11-14

**Authors:** Cynthia Jinno, Perot Saelao, Elizabeth A. Maga, Annie King, Dan Morash, Steve Zicari, Xiang Yang, Yanhong Liu

**Affiliations:** 1Department of Animal Science, University of California, Davis, CA 95616, USA; cnjinno@ucdavis.edu (C.J.); eamaga@ucdavis.edu (E.A.M.); ajking@ucdavis.edu (A.K.); 2United States Department of Agriculture, Agricultural Research Service, The Honey Bee Breeding, Genetics, and Physiology Research, Baton Rouge, LA 70820, USA; Perot.Saelao@usda.gov; 3California Safe Soil, LLC, McClellan, CA 95652, USA; dan.morash@calsafesoil.com (D.M.); steve.zicari@calsafesoil.com (S.Z.)

**Keywords:** enzymatically digested food waste, fatty acid profile, fecal microbiota, growing-finishing pigs, meat quality

## Abstract

**Simple Summary:**

Food waste has been negatively impacting the environment, which can harm the human population. Enzymatic digestion is a great way to reuse and recycle food waste, and its product could be used to feed growing-finishing pigs. In this experiment, we investigated the meat quality and the fecal microbiota of pigs fed with enzymatically digested food waste. Results indicate feeding 100% enzymatically digested food waste did not alter the meat quality of finishing pigs in comparison to the pigs fed with traditional corn-soybean meal diet. However, pigs fed with enzymatically digested food waste contained more omega-3 fatty acids (i.e., eicosapentaenoic acid and docosahexaenoic acid) in the back-fat than pigs fed with corn-soybean diet. Moreover, feeding enzymatically digested food waste remarkably impacted fecal microbiome diversity of pigs, particularly increased the relative abundance of *Lachnospiraceae* family that was suggested to be positively correlated with the concentrations of beneficial fatty acids in the host. In summary, feeding enzymatically digested food waste to growing-finishing pigs not only contributes to the sustainability of agriculture, but also provides more beneficial fatty acids to pork consumers.

**Abstract:**

This experiment aimed to evaluate meat quality, fatty acid profile in back-fat, and fecal microbiota of growing-finishing pigs fed with liquid enzymatically digested food waste. Fifty-six crossbred pigs (approximately 32.99 kg body weight) were assigned to one of two treatments with seven replicate pens and four pigs per pen. Pigs were fed with control (corn-soybean meal diets) or food waste from d 0 to 53, while all pigs were fed with the control diet from d 53 to 79. The 16S rRNA sequencing was used to analyze microbiota of feces collected on d 0, 28, 53, and 79. Meat quality and carcass characteristics were measured in one pig per pen at the end of the experiment. Pigs fed with food waste contained more (*p* < 0.05) eicosapentaenoic acid (EPA) and docosahexaenoic acid (DHA) in back-fat. Feeding food waste increased (*p* < 0.05) the relative abundances of *Lachnospiraceae* and *Ruminococcaceae*, but decreased (*p* < 0.05) the relative abundances of *Streptococcaceae* and *Clostridiaceae* in feces on d 29 or d 53. In conclusion, feeding enzymatically digested food waste did not affect pork quality, but provided more beneficial fatty acids to pork consumers and altered the fecal microbiota in growing-finishing pigs.

## 1. Introduction

According to the Food and Agriculture Organization [1], food waste is defined as a safe and nutritious food that has been discarded before human consumption. In 2015, approximately 22% of the U.S. municipal solid waste in landfills was food waste, which exceeded the percentages of other wastes, including paper, plastics, and textiles [2]. The accumulation of food waste in landfills may negatively impact the environment by releasing methane gas [3,4]. Several methods have been widely applied to reduce or recycle food waste, including composting, anaerobic digestion, and enzymatic digestion [5,6,7]. Enzymatic digestion can break down macronutrients (carbohydrates, proteins, and fats) to highly digestible nutrients. The current research focused on the food waste product produced through enzymatic digestion. The detailed procedures of food waste processing have been described in our previously published research [7,8]. Briefly, food waste (fruits, vegetables, meat, and bakery products) collected from local supermarkets were mixed and digested with enzymes, and then pasteurized 30 min at 75 to 77 °C. The final product was verified free of foodborne pathogens and was named enzymatically digested food waste in the current experiment. The chemical composition of enzymatically digested food waste indicated this food waste product contains a balanced amino acid profile for growing pigs [7]. Our previous study has also reported that pigs fed with enzymatically digested food waste had a similar fed efficiency to that pigs fed traditional corn-soybean meal diet [7]. However, carcass characteristics and meat quality of finishing pigs fed with this food waste product were not evaluated.

It has been widely accepted that diets or nutrients have remarkable impacts on the gut microbiota of pigs and human beings in different physiological stages [9,10]. The modification of gut microbes could reversely affect metabolic activities of the hosts [11,12]. However, there are limited research connecting nutrients, gut microbiota, and meat quality of pigs. Yan et al. [13] have stated that fiber (i.e., inulin) supplementation remarkably changed gut bacteria community by increasing the relative abundance of *Bacteroides* in cecal content of growing pigs. This change may consequently impact the types and concentrations of microbial metabolites, leading to reduced back fat accumulation in pigs [13]. Moreover, it has been recently reported that gut microbiota in obese animals may facilitate fat deposition, impacting body composition and meat quality [14]. In the present experiment, the enzymatically digested food waste contains more fat (30.23% on a dry matter basis), compared with traditional corn and soybean meal diets (approximately 5–6% fat on a dry matter basis) [7]. Therefore, there were two objectives in this study: (1) To evaluate the effects of feeding the enzymatically digested food waste on carcass characteristics and meat quality of finishing pigs; (2) to determine the impacts of feeding food waste on the fecal microbiota of growing-finishing pigs.

## 2. Materials and Methods

The animal use protocol was reviewed and approved by the Institutional Animal Care and Use Committee (#19322) at the University of California, Davis (UC Davis). The animal trial was conducted at the Swine Teaching and Research Center at UC Davis.

### 2.1. Animals, Husbandry, Experimental Design, and Dietary Treatments

The detailed procedures for the animal trial have been described in our previously published research [7], including animal husbandry, experimental design, and diets. Briefly, 56 crossbred growing pigs (approximately 32.99 kg initial body weight (BW)) were group housed in 14 concrete pens (2.44 × 3.81 m^2^) with two barrows and two gilts per pen. Pigs were randomly assigned to one of two treatments (control vs. food waste) with seven replicate pens per treatment. The experimental design was a randomized complete block design with weight within litter as blocks and pen as the experimental unit. Each pen contained a nipple drinker and a dry feeder to allow pigs having free access to water and feed.

The animal trial was conducted 79 days with 28 days as phase 1 (grower diets), 25 days as phase 2 (finisher-1 diets), and 26 days (finisher-2 diet) as phase 3. Pigs were fed either control or food waste diet in phases 1 and 2, and all pigs were fed with the control diet in phase 3. Control diets were formulated based on corn and soybean meal (Table 1). A liquid diet was enzymatically digested food waste supplemented with vitamin-mineral premix and salt. Liquid diet was weekly prepared and provided by California Safe Soil, LLC. (McClellan, CA, USA). All diets met current estimates for nutrient requirements for growing and finishing pigs [15].

The subsamples of diets were collected weekly and thoroughly mixed with analyzing dry matter (Method 930.15 [16]), crude fat (Method 954.02 [16]), and fatty acid profile by gas-liquid chromatography according to Methods 965.49 and 996.06 [16]. At the conclusion of the animal trial, 14 pigs (1 pig per pen) with BW close to the average BW of the pen were selected and harvested at the Meat Laboratory of UC Davis under the supervision of a federal inspector.

### 2.2. Carcass Characteristics and Fresh Meat Quality

#### 2.2.1. Hot Carcass Weight (HCW), Estimated Lean, and Back-Fat

The HCW was obtained 45-min postmortem. Carcass yield was calculated by dividing HCW by final live weight. Carcasses were chilled at 4 °C for approximately 24 h. The left side of each chilled carcass was split between the 10th and 11th rib to expose the longissimus muscles (LM) for measuring loin eye area (LEA) and back-fat, according to the procedures described by Little et al. [17]. Back-fat samples were also analyzed for dry matter, crude fat, and fatty acid profile using the same methods described above. Estimated carcass lean was calculated according to the equation from Boler et al. [18]:Estimated carcass lean, % = [8.588 + (0.465 × HCW, lb.) − (21.896 × 10th rib back-fat, in.) + (3.005 × 10th rib LEA, in.2)] ÷ HCW, lb.(1)

#### 2.2.2. Water Holding Capacity and Objective Tenderness

One random 1.25-cm chop and three slices of 2.54-cm thick chops were excised from the posterior of the 10th rib to analyze drip loss and objective tenderness. To measure drip loss, the chops were placed in individual bags and suspended with a metal hook for 24 h at 4 °C [19]. All chops were weighed before and after suspension to calculate drip-loss as the percentage of weight loss in each chop. The objective tenderness and cook loss were analyzed by following the procedures of the American Meat Science Association [20]. Briefly, the 2.54-cm thick chops were vacuum packaged and frozen at −20 °C until analysis. Prior to analysis, the chops were completely thawed in a cooler at 4 °C for approximately 24 h. Thawed chops were trimmed to 0.3 cm subcutaneous fat and weighed, before cooked on Weber Q 100 Gas Grill (Weber-Stephen Products Co., Palatine, IL, USA). Chops were grilled on one side until they reached an internal temperature of 40 °C, flipped and cooked to a final internal temperature of 71 °C (Fluke 52 K/J Two Channel Digital Thermometer, Fluke Co., Everett, WA, USA). Cooked chops were immediately weighed after cooking to calculate cook loss, which was reported as moisture loss during cooking as a percentage of raw weight. The cooked chops were then chilled overnight at 4 °C. A metal probe was used to remove four cores (1.27 cm diameter) from the center of each chop perpendicular to the muscle fiber, and then the cores were sheared using a Warner-Bratzler Shear Force equipment (WBSF; Salter, G-R Electric, Manhattan, KT, USA). The average WBSF values of four cores were reported for each chop.

#### 2.2.3. Subjective Color, Marbling Score, Firmness Score, pH, and Moisture Content

Subjective color, marbling score, and firmness score were measured according to the standards established by the National Pork Producers Council [21,22]. Objective color of the lean muscle of chops was measured using a portable spectrophotometer (MiniScan EZ 4500L; Hunter Associates Laboratory Inc., Reston, VA, USA) utilizing an illuminant D65 light source, a 10° observer, and an aperture size of 25 mm. The results were expressed as L* (lightness), a* (redness), and b* (yellowness) values. To analyze pH, individual thin chops (~10 g) were placed in a Whirl-Pak bag and homogenized with deionized water in a 1:5 ratio by weight using a masticator (IUL masticator, IUL, S.A., Barcelona, Spain) for 2 min. Ultimate pH of the homogenate was measured with a hand-held pH meter (VWR Benchtop Meters, Radnor, PA, USA; 2 point calibration: pH 4 and 7). To analyze moisture content, 5 g of the sample was trimmed to 0 cm subcutaneous fat, placed into aluminum pans, covered with Whatman #1 filter paper, and oven dried at 110 °C for 24 h. Weight loss was measured to calculate moisture content.

### 2.3. Carcass Composition

The right sides of the carcasses were chilled approximately 24 h after slaughter, and fabricated. Shoulders and hams were fabricated according to the methods from Little et al. [17] and Boler et al. [18]. The fabricated cuts were weighed to calculate the following:Lean cutting yield, % = [(trimmed ham, kg + trimmed loin, kg + Boston butt, kg + picnic, kg) ÷ half side chilled carcass weight, kg] × 100(2)

Boneless lean cutting yield, % = [(inside ham, kg + outside ham, kg + knuckle, kg + light butt, kg + shank, kg + Canadian back, kg + tenderloin, kg + sirloin, kg + boneless Boston butt, kg + boneless picnic, kg) ÷ half side chilled carcass weight, kg] × 100(3)

Carcass cutting yield, % = [(trimmed ham, kg + trimmed loin, kg + Boston butt, kg + picnic, kg + trimmed belly, kg) ÷ half side chilled carcass weight, kg] × 100(4)

Boneless carcass cutting yield, % = [(inside ham, kg + outside ham, kg + knuckle, kg + light butt, kg + shank, kg + Canadian back, kg + tenderloin, kg + sirloin, kg + boneless Boston butt, kg + boneless picnic, kg + trimmed belly, kg) ÷ half side chilled carcass weight, kg] × 100(5)

### 2.4. Fecal Microbiota

Fecal samples were collected from the same pig per pen via anal stimulation at the beginning of the experiment, on d 28, d 53, and d 79. Fresh fecal samples were immediately stored in −80 °C prior to further analysis. Bacterial DNA was extracted from fecal samples using the Quick-DNA Fecal/Soil Microbe Kit (Zymo Research, Irvine, CA, USA) following the manufacturer’s instructions. Extracted bacterial DNA was amplified with PCR, targeting the V4 region of the 16S rRNA gene with primers 515 F (5′-XXXXXXXXGTGTGCCAGCMGCCGCGGTAA-3′) with an 8 bp barcode (X) and Illumina adapter (GT) and 806 R (5′-GGACTACHVGGGTWTCTAAT-3′) [23]. Amplification included thermocycling conditions of 94 °C for 3 min for denaturation, 35 cycles of 94 °C for 45 s, 50 °C for 1 min, 72 °C for 1.5 min, and 72 °C for 10 min (final elongation). To reduce polymerase chain reaction (PCR) bias, each sample was amplified in triplicate. Each PCR reaction included 2 µL of template DNA, 0.5 µL of barcode primer, 0.5 µL (10 µM) of reverse primer, 12.5 µL of GoTaq 2X Green Master Mix (Promega, Madison, WI, USA), and 9.5 µL of nuclease free water. The triplicate PCR products were pooled and subjectively quantified based on the brightness of the bands on a 2% agarose gel with SYBR safe (Invitrogen Co., Carlsbad, CA, USA). All amplicons were then pooled at equal amounts. The pooled library was purified using the QIAquick PCR Purification Kit (Qiagen, Hilden, Germany) and submitted to the UC Davis Genome Center DNA Technologies Core for 250 bp paired-end sequencing on the Illumina MiSeq platform (Illumina, Inc. San Diego, CA, USA).

The software sabre (https://github.com/najoshi/sabre) was used to demultiplex and remove barcodes from raw sequences. Sequences were then imported into Quantitative Insights Into Microbial Ecology 2 (QIIME2; version 2018.6) for downstream filtering and bioinformatics analysis [24,25]. Plugin q2-dada2 [26] was used for quality control and constructing features. Taxonomic classification was assigned using the feature-classifier plugin [27] trained with SILVA rRNA database 99% Operational Taxonomic Units (OTU) (version 132; [28]).

### 2.5. Statistical Analysis

For all data except for fecal microbiota, normality was verified, and outliers were identified using the UNIVARIATE procedure (SAS Institute Inc., Cary, NC, USA). Data were analyzed by SAS ANOVA using PROC MIXED in a randomized complete block design with pen as the experimental unit. The statistical model included diet as the main effect and blocks (replicate and group) as random effects. Statistical significance and tendency were considered at *p* < 0.05 and 0.05 ≤ *p* < 0.10, respectively.

Data visualization and statistical analysis for fecal microbiota were conducted using the R program (version 3.6.1; [29]). Two alpha diversity indices, Chao1 and Shannon, were calculated using the phyloseq package [30]. Beta diversity was calculated based on the Bray-Curtis dissimilarity for principal coordinates analysis (PCoA). The homogeneity of multivariate dispersions was tested by the vegan package [31] using the betadisper function, before the adonis function was used to calculate PERMANOVA with 999 replicate permutations [32]. Pairwise post hoc comparison was performed using the pairwiseAdonis package [33]. Relative abundance was calculated using the phyloseq package and visualized using ggplot2 package in R [34].

## 3. Results and Discussion

### 3.1. Carcass Characteristics and Meat Quality

The complete nutrient composition of enzymatically digested food waste, including amino acids, carbohydrates, minerals, and fatty acids profile has been published in Jinno et al. [7]. The data of growth performance was also published in Jinno et al. [7]. In brief summary, growing-finishing pigs fed with food waste diet as the sole ingredient grew slower than pigs fed a corn and soybean meal diet. Pigs in the food waste treatment had lighter final BW than pigs in control, although those pigs had better feed efficiency at the last phase of the experiment. Therefore, the pigs fed with food waste diet had lighter (*p* = 0.05) ending live weight (Table 2). Pigs fed with food waste also had lower HCW (*p* < 0.05) than the pigs fed with control. No differences were observed in carcass yield, loin eye area, 10th rib back-fat, and estimated carcass lean.

No differences were also observed in LM moisture content, shear force values, cook loss, pH, objective color (L*, a*, and b*), subjective color, marbling score, and drip loss between two dietary treatments (Table 3). Pork chops from pigs fed with food waste tended to have lower (*p* = 0.087) subjective firmness score than those from pigs fed with control. Consistent with ending live weight, the half carcass chilled weight also tended to be lighter (*p* = 0.054) in pigs fed with food waste, compared with pigs in the control treatment (Table 4). However, no differences were observed in lean cutting yield, boneless lean cutting yield, carcass cutting yield, and boneless carcass cutting yield between two treatments.

There are limited data reporting carcass characteristics, and meat quality of pigs fed diets containing or consisting solely of food waste. Chae et al. [35] and Kjos et al. [36] reported that feeding up to 40% of dried kitchen waste did not affect carcass characteristics of finishing pigs. Westendorf et al. [37] also revealed that feeding cafeteria food waste (22.4% dry matter) had similar pork quality compared with pigs fed corn and soybean meal. However, Choe et al. [38] pointed out that pork from pigs consumed dried food waste collected from South Korean restaurants had greater L* and b* values, indicating a brighter and yellower appearance. In the same study, a higher drip loss was also detected in the pork from pigs fed with dried food waste [38]. The authors [38] explained this pale, soft, and exudative meat was likely due to the lower ultimate LM pH in pork from pigs fed with food waste [39,40]. Considering the high fat concentration in the enzymatically digested food waste, pigs in the present study were only fed the food waste diet at growing and early finishing stages (phases 1 and 2) to avoid soft fat in pork. Feeding food waste to growing-finishing pigs did not affect carcass characteristics, LM quality, and carcass cutability, in comparison to feeding corn and soybean meal. Those observations are consistent with the majority of published research described above. Thus, feeding food waste containing high fat did not negatively impact the meat quality of finishing pigs if they were only fed at growing and early finishing periods.

### 3.2. Back-Fat Composition and Fatty Acids Profile

Dietary lipids are important energy sources for animal maintenance, growth, and production. Thus, increasing the concentration of dietary fat/lipids is a practical method to enhance the growth rate and feed efficiency of pigs [41]. However, adding fat to diets may also increase carcass fatness and reduce carcass leanness [42]. In the present study, no differences were observed in moisture and crude fat content in the back-fat of pigs between control and food waste treatments (Table 5). In combination with the results of estimated carcass lean percentage described above, the findings of this study indicate feeding enzymatically digested food waste in growing, and early finishing stages did not increase carcass fatness.

Feeding food waste as the sole diet significantly (*p* < 0.05) impacted the fatty acid profile of back-fat, although all pigs were fed control in the last phase of this experiment (Table 5). The back-fat in pigs fed with food waste had greater (*p* < 0.05) concentration of pentadecylic acid, margaric acid, myristoleic acid, palmitoleic acid, oleic acid, vaccenic acid, gondoic acid, eicosapentaenoic acid (EPA), and docosahexaenoic acid (DHA), compared with the back-fat of pigs in the control group. However, pigs fed with food waste contained less (*p* < 0.05) palmitic acid, arachidic acid, linoleic acid, linolenic acid, and eicosadienoic acid in back-fat, compared with pigs fed control diet. Control diets contained 16.81% to 16.95% of saturated fatty acids and 81.53 to 82.12% of unsaturated fatty acids in overall crude fat. Food waste diet contained 36.36% of saturated fatty acids and 54.61% of unsaturated fatty acids in overall crude fat. However, control pigs contained more (*p* < 0.05) saturated fatty acids (30.64 vs. 27.25%) in their back-fat than pigs fed with food waste diet. There was no difference in the concentration of total unsaturated fatty acids in the back-fat of pigs fed different diets.

Fatty acids accumulated in pork fat can be obtained from direct absorption of fatty acids from the diet and from de novo lipogenesis of non-lipid substrates. In the finishing phase, at least half of the daily lipid deposition is associated with de novo lipogenesis when pigs were fed with a typical corn and soybean meal diet [43]. The increased de novo lipogenesis may affect fatty acid compositions in pork fat by increasing the proportion of saturated fatty acids in pork fat [44]. In addition, it has been reported that increased dietary fat content and dietary palmitic acid concentration could inhibit de novo lipogenesis in pigs [45,46]. Therefore, (1) the greater amount of saturated fatty acids in back-fat of pigs fed with the control diet occurred, due to de novo lipogenesis of glucose liberated from corn starch, and (2) pigs fed with food waste may have had more dietary fatty acids deposited to pork fat. This is also the likely reason for the increased palmitic acid in back-fat of pigs fed with the control diets.

As discussed above, the dietary fatty acid composition could partially reflect the fatty acid profile in pork fat, and vice versa. The food waste diet was produced by enzymatically digesting supermarket food wastes, which included approximately 20% of meat from livestock or fish. Thus, the back-fat in pigs fed with food waste diet were composed of more monosaturated fatty acids and polyunsaturated fatty acids, including oleic acid, EPA, and DHA than pigs in control. Although there are apparent health benefits with consumption of polyunsaturated fatty acids, the greater degree of unsaturation of fatty acids in meat may also increase lipid peroxidation [47,48,49]. In the present study, pigs provided food waste had a greater concentration of monounsaturated fatty acids, DHA, and EPA, but less total polyunsaturated fatty acids than pigs fed with corn and soybean meal diet. It has been reported that linoleic acid greater than 15% of total fat may result in soft fat in pork [50]. Results of our work revealed that fat in pigs fed with food waste diet contained much lower linoleic acid (15.08 vs. 19.51%) compared with pigs fed with the control diet. Therefore, current results suggested that feeding food waste had the potential to enhance beneficial fatty acid content in pork.

### 3.3. Fecal Microbiota

Alpha diversity indices of fecal microbiota are shown in Figure 1. Chao1 and Shannon indices of fecal microbiota were decreased (*p* < 0.05) from d 0 to d 28 and plateaued throughout the feeding program when pigs fed with the control diet. However, Chao1 index of fecal microbiota was the greatest (*p* < 0.05) on d 28, but the lowest (*p* < 0.05) on d 79 for pigs fed with food waste. Shannon index of fecal microbiota was increased (*p* < 0.05) on d 28 then remained stable on d 53, but reduced (*p* < 0.05) in fecal samples on d 79 when pigs were fed with food waste. Chao1 index measures microbial population richness and Shannon index measures microbial diversity (richness and evenness) of fecal microbiota [51,52]. Results of alpha diversity indicated that feeding enzymatically digested food waste increased the richness and evenness of microbial community in feces.

Total reads were filtered to remove the bacterial phyla with low prevalence and abundance, and taxa present in less than 50% of the samples. A total of 797,905 qualified reads were obtained with a mean of 13,259 reads per sample. Interactive effect of diet and day was observed in the overall bacterial composition (Adonis, *p* < 0.05) with statistically no heterogeneous dispersion (Betadisper, df = 7, *F* = 0.9621, *p* = 0.4693). Differences were observed in the bacterial composition at phyla level of fecal microbiota between control and food waste on d 28 (Pairwise-Adonis, R^2^ = 0.44, *p* < 0.05) and d 53 (Pairwise-Adonis, R^2^ = 0.32, *p* < 0.05). Pigs fed with food waste formed separate clusters from pigs fed with control (Figure 2). This observation indicated that bacterial composition was altered when pigs were fed with enzymatically digested food waste. However, all pigs were clustered together on d 79, indicating pigs in the food waste treatment had similar fecal bacterial composition compared with pigs in the control treatment at the end of the experiment.

The three dominant phyla in fecal microbiota were Firmicutes, Bacteroidetes, and Proteobacteria, regardless of treatments and dates (Figure 3). No difference was observed in the relative abundances of Proteobacteria between control and food waste treatments. However, pigs fed with food waste had lower (*p* < 0.05) relative abundance of Firmicutes, but higher (*p* < 0.05) relative abundance of Bacteroidetes than pigs in control on d 28. No difference was observed in the relative abundance of those two phyla on d 0, 53, and 79. The relative abundance of Euryarchaeota was higher (*p* < 0.05) on d 53, and the relative abundance of Actinobacteria was higher (*p* < 0.05) on d 28 and 53 in fecal samples collected from pigs fed with food waste than pigs in control. The increase in the relative abundance of Actinobacteria was most likely due to the high fiber content of food waste diet (84.2 g/kg acid detergent fiber), compared with the control diets (44.5 and 47.0 g/kg acid detergent fiber in phases 1 and 2, respectively). This result was consistent with gut microbiota changes in humans when a high fiber diet was consumed [53]. No difference was observed in Firmicutes to Bacteroidetes ratio between two dietary treatments, but the ratio was increased (*p* < 0.05) as the age of animal was increased (Figure 4). The Firmicutes to Bacteroidetes ratio has been consistently reported as an important microbial marker for obesity in human studies, as the ratio is increased in obese people compared with lean people [54,55]. Therefore, the increased Firmicutes to Bacteroidetes ratio in feces was likely associated with the enhanced fat accumulation in the body as the age of pigs increased.

Within the Firmicutes at the family level, feeding enzymatically digested food waste increased (*p* < 0.05) the relative abundances of *Acidaminococcaceae* (0.75% vs. 0.22%), *Lachnospiraceae* (41.68% vs. 17.30%), and *Ruminococcaceae* (26.71% vs. 11.38%), but decreased (*p* < 0.05) the relative abundances of *Clostridiaceae* (5.90% vs. 21.06%), *Lactobacillaceae* (1.66% vs. 5.53%)*,* and *Strepococcaceae* (5.36% vs. 19.95%) in pig fecal samples collected on d 28 (Figure 5). Feeding enzymatically digested food waste also increased (*p* < 0.05) the relative abundances of *Lachnospiraceae* (23.92% vs. 10.60%) and *Peptostreptococcaceae* (28.86% vs. 16.77%) and decreased (*p* < 0.05) the relative abundances of *Clostridiaceae* (22.80% vs. 32.45%) and *Strepococcaceae* (0.11% vs. 15.24%) in pig fecal samples collected on d 53, compared with control.

Growing evidence indicates the significant impacts of nutrients on gut microbiota in pigs and in humans [9,10]. However, many other internal and external factors may affect gut microbiota diversity as well, such as animal health status, environment, genetics, etc. In the current study, two major differences existed between the two treatments, including nutrient compositions and diet form. The control diets were a solid diet based on corn and soybean meal, whereas, food waste diet was liquid with a relatively high concentration of fat. Thus, it was highly possible that these two external factors had co-founding effects on the gut microbiota of growing-finishing pigs, which needs to be investigated in future research. To our knowledge, there was no research exploring the impacts of liquid feeding on gut microbiota of growing pigs. The following discussion mainly focused on the impacts of nutrients in experimental diets on gut microbiota. Results in the current study were in close agreement with the findings reported in few published research with adult humans and rats [56,57,58]. For instance, a rat study reported by Zhao et al. [56] demonstrated that high fat diet (45% fat) suppressed the relative abundance of *Lachnospiraceae* and *Acidaminococcaceae* in fecal contents of rats, compared with normal fat diet (10% fat). A reduced relative abundance of *Lactobacillaceae* and an increased relative abundance of *Ruminococcaceae* was also observed in the feces of rats fed with high fat (43% fat) diet, compared with rats fed with normal diet (12% fat) [57]. In addition, Costantini et al. [58] revealed that feeding omega-3 fatty acids increased abundance of the *Lachnospiraceae* members. Therefore, the increased relative abundance of *Lachnospiraceae* in feces that was observed in this study was likely due to the relatively high concentration of EPA and DHA in food waste.

Limited research has been reported on the impacts of food waste on the gut microbiome of pigs, in particular, in the growing and finishing stages. Results from the current experiment indicated that the nutrient composition (high fat) in a complete feed has significant effects on the gut microbiome of growing-finishing pigs. More research is needed to decipher the interactions between nutrients and the gut microbiome and their potential regulatory roles in animal health and overall production. Interestingly, the differences of fecal microbiota between treatments disappeared, when all pigs were switched to control diet in the last phase of the experiment. This observation also confirmed that nutrients play huge roles in regulating gut microbiota diversity. This was also the likely reason that feeding enzymatically digested food waste in growing, and early finishing stages did not affect meat quality and carcass characteristics of finishing pigs.

## 4. Conclusions

Feeding enzymatically digested food waste to growing and early finishing pigs did not affect meat quality, but did change the back-fat fatty acid profile. There were more beneficial fatty acids, such as EPA and DHA, in back-fat of pigs fed food waste diet. The shelf-life of pork from pigs fed food waste should be further investigated because the increased amount of EPA and DHA may affect the shelf life of pork by changing the rate of lipid oxidation. Feeding enzymatically digested food waste also modified the fecal microbiota of growing-finishing pigs. Changes in fecal microbiota are most likely due to the differences in nutrient compositions of diets containing food waste and corn-soybean meal. Future studies are warranted to further analyze the fecal microbiome of growing pigs, and probably extending to weaned pigs when they are fed with enzymatically digested food waste as dry form.

## Figures and Tables

**Figure 1 animals-09-00970-f001:**
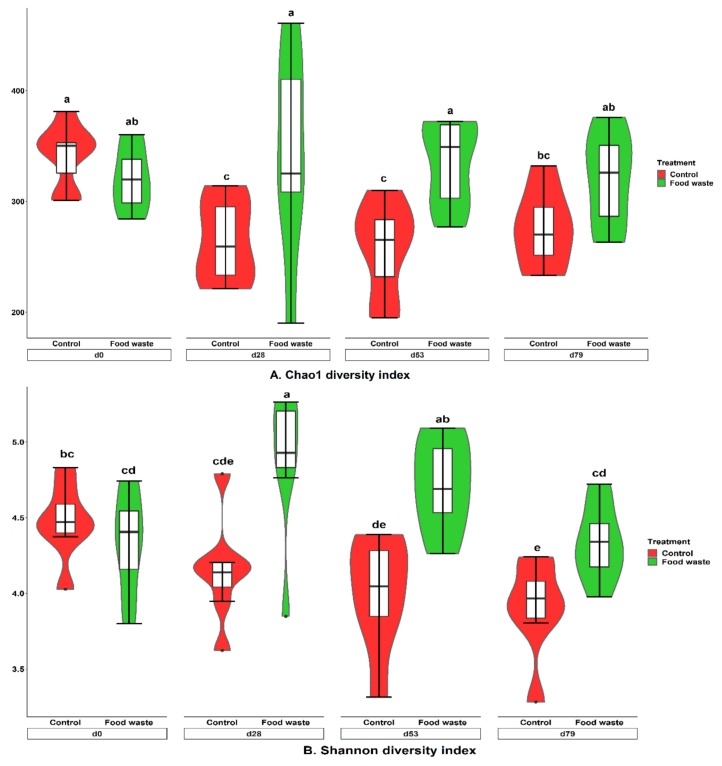
Pig fecal alpha diversity as indicated by Chao1 (**A**) and Shannon (**B**). For Chao1 index, Diet, *p* < 0.001; Day, *p* = 0.20; Diet × day, *p* < 0.05. For Shannon index, Diet, *p* < 0.001; Day, *p* < 0.05; Diet × day, *p* < 0.01. ^a–e^ Means without a common superscript are different (*p* < 0.05).

**Figure 2 animals-09-00970-f002:**
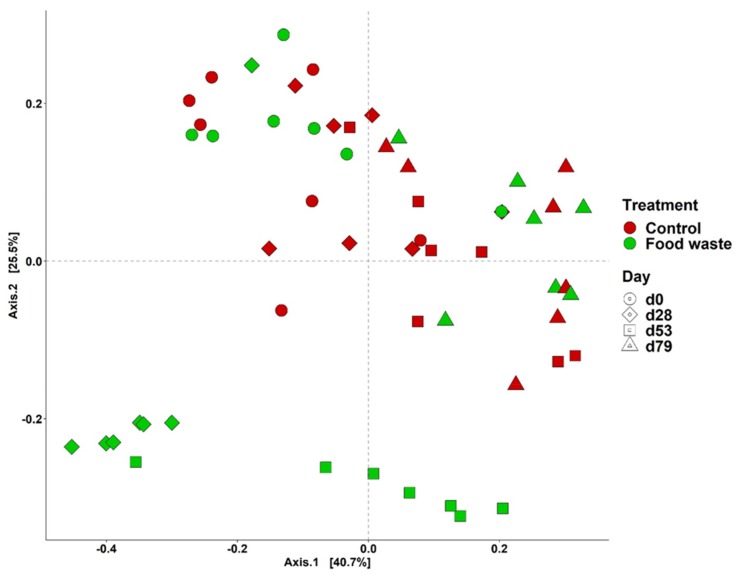
Beta diversity of dietary treatments and different phases were analyzed by principal coordinate analysis (PCoA) based on the Bray-Curtis dissimilarity. Symbols indicate dietary treatments and colors indicate different dates.

**Figure 3 animals-09-00970-f003:**
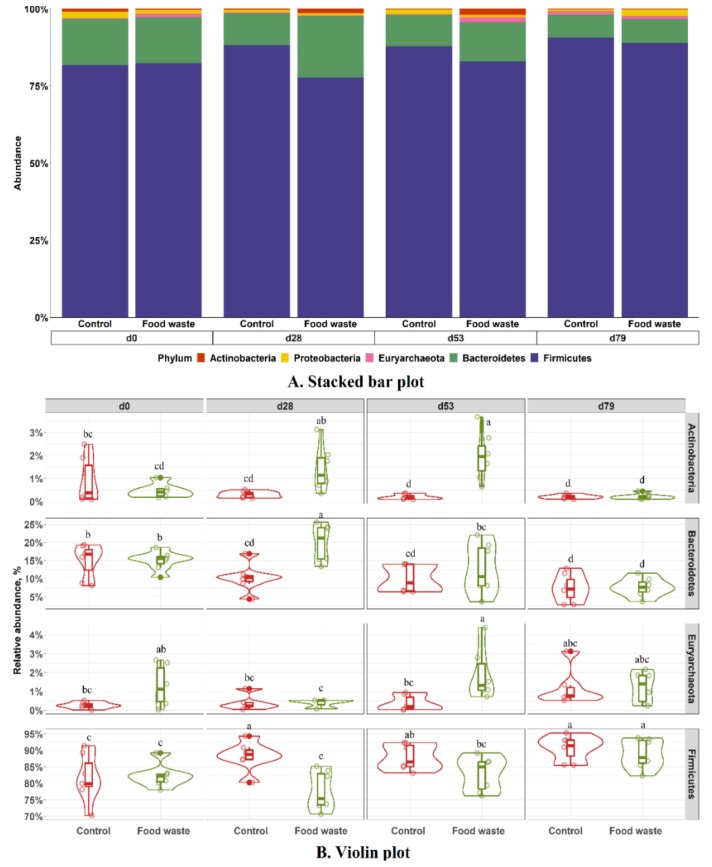
Stacked bar plot showing the relative abundance of bacterial phyla in feces of pigs fed with the control diet or food waste diet (**A**). Violin plot showing the relative abundance of individual bacterial phylum in feces of pigs (**B**). Diet, *p* < 0.05; Day, *p* < 0.05; Diet × day, *p* < 0.05. ^a–d^ Means without a common superscript are different (*p* < 0.05).

**Figure 4 animals-09-00970-f004:**
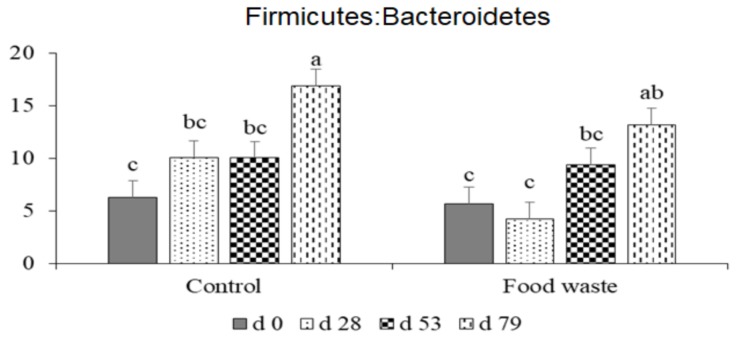
Firmicutes to Bacteroidetes abundance ratio in fecal samples of growing-finishing pigs fed with control or food waste diet. ^a–c^ Least square means without a common superscript are different (*p* < 0.05).

**Figure 5 animals-09-00970-f005:**
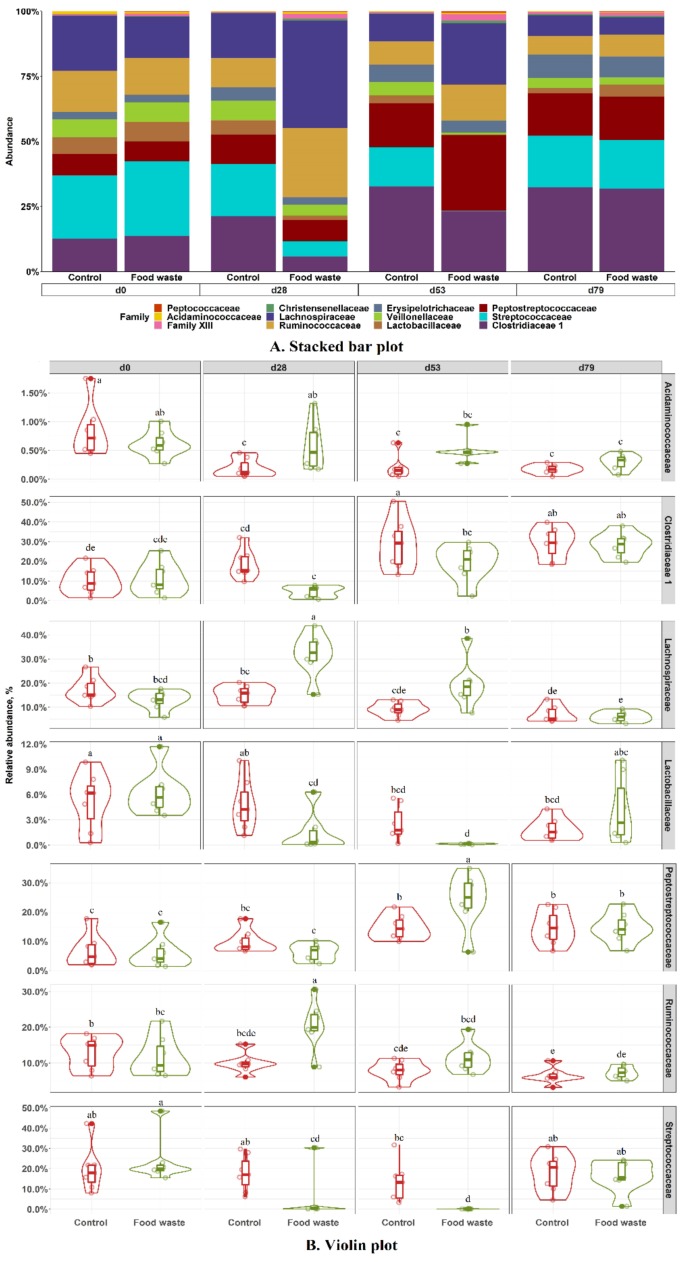
Stacked bar plot of relative abundance of Firmicutes family in feces of pigs fed with the control diet or food waste diet (**A**). Violin plot showing the relative abundance of selected individual Firmicutes family in feces of pigs (**B**). Diet, *p* < 0.05; Day, *p* < 0.05; Diet × day, *p* < 0.05. ^a–e^ Means without a common superscript are different (*p* < 0.05).

**Table 1 animals-09-00970-t001:** Ingredient composition of control diets and liquid feed.

Ingredient, g/kg	Control			100% Food Waste ^1^
	Phase 1	Phase 2	Phase 3	
Ground corn	682.0	743.7	777.0	−
Soybean meal, 48% crude protein	270.0	210.0	180.0	−
Soybean oil	20.0	20.0	20.0	−
Limestone	8.3	8.0	7.5	−
Dicalcium phosphate	10.5	9.0	7.0	−
L-Lysine HCL, 78% Lysine	1.8	2.0	1.4	−
DL-Methionine	0.2	−	−	−
L-Threonine	0.2	0.3	0.1	−
Food waste	−	−	−	998.2
Salt	4.0	4.0	4.0	1.0
Vitamin-mineral premix ^2^	3.0	3.0	3.0	0.8
Total	1000.0	1000.0	1000.0	1000.0
Analyzed nutrients, dry matter basis, g/kg				
Dry matter	863.4	867.3	880.2	237.2
Crude protein	212.1	173.7	158.9	257.4
Crude fat	50.4	53.0	58.1	302.3
Acid detergent fiber	44.5	47.0	47.9	84.2
Neutral detergent fiber	124.5	107.0	112.3	113.5
Saturated fatty acid, % of crude fat				
Myristic (14:0)	0.08	0.07	0.06	2.48
Pentadecylic (15:0)	0.02	0.02	0.02	0.40
Palmitic (16:0)	12.88	12.77	12.81	21.82
Margaric (17:0)	0.11	0.11	0.11	0.97
Stearic (18:0)	3.03	2.88	3.00	10.34
Arachidic (20:0)	0.38	0.35	0.35	0.20
Behenoic (22:0)	0.25	0.25	0.26	0.10
Total	16.95	16.63	16.81	36.36
Unsaturated fatty acid, % of crude fat				
Myristoleic (14:1)	0.00	0.00	0.00	0.45
Palmitoleic (16:1)	0.16	0.13	0.11	3.45
Oleic (18:1)	23.01	23.58	23.71	34.48
Vaccenic (18:1)	1.22	1.17	1.20	2.19
Linoleic (18:2)	52.43	52.85	52.14	10.95
Linolenic (18:3)	4.31	4.01	4.09	1.36
Gondoic (20:1)	0.35	0.33	0.33	0.44
Eicosadienoic (20:2)	0.02	0.02	0.03	0.18
Arachidonic (20:4)	0.00	0.00	0.00	0.34
Eicosapentaenoic (EPA, 20:5)	0.00	0.00	0.00	0.34
Docosaherxaenoic (DHA, 22:6)	0.00	0.00	0.00	0.28
Monounsaturated fatty acids	24.77	25.24	25.39	41.10
Polyunsaturated fatty acids	56.76	56.89	56.25	13.52
Total fatty acid, % of crude fat	81.53	82.12	81.64	54.61

^1^ Vitamin-mineral premix (approximately 0.3% as dry matter basis) and salt (approximately 0.4% as dry matter basis) were added to the food waste diet during preparation.^2^ Provided the following quantities of vitamins and micro-minerals per kilogram of complete diet: Vitamin A as retinyl acetate, 11,128 IU; vitamin D_3_ as cholecalciferol, 2204 IU; vitamin E as _DL_-α-tocopheryl acetate, 66 IU; vitamin K as menadione nicotinamide bisulfite, 1.42 mg; thiamin as thiamine mononitrate, 0.24 mg; riboflavin, 6.58 mg; pyridoxine as pyridoxine hydrochloride, 0.24 mg; vitamin B_12_, 0.03 mg; _D_-pantothenic acid as _D_-calcium pantothenate, 23.5 mg; niacin as nicotinamide, 1.0 mg, and nicotinic acid, 43.0 mg; folic acid, 1.58 mg; biotin, 0.44 mg; Cu, 10 mg as copper sulfate; Fe, 125 mg as iron sulfate; I, 1.26 mg as potassium iodate; Mn, 60 mg as manganese sulfate; Se, 0.3 mg as sodium selenite; and Zn, 100 mg as zinc oxide.

**Table 2 animals-09-00970-t002:** Carcass characteristics of finishing pigs fed control and enzymatically digested food waste.

Item ^1^	Control	Food Waste	SEM	*p*-Value
Ending live weight, kg	105.79	97.86	3.36	0.050
HCW, kg	84.11	76.98	2.91	0.028
Carcass yield, %	79.57	78.70	0.93	0.320
Loin eye area, cm^2^	49.25	47.45	3.079	0.444
10th rib back-fat, cm	2.34	1.76	0.244	0.152
Estimated carcass lean, %	52.65	55.57	1.344	0.186

^1^ Each least squares mean represents seven observations. HCW = hot carcass weight, Estimated carcass lean, % = [8.588 + (0.465 × HCW, lb.) − (21.896×10th rib back-fat, in.) + (3.005×loin eye area, in.^2^)] ÷ HCW, lb. × 100 (Burson and Berg, 2001).

**Table 3 animals-09-00970-t003:** Characteristics of longissimus muscle of finishing pigs fed control and enzymatically digested food waste.

Item ^1^	Control	Food Waste	SEM	*p*-Value
Moisture, %	74.28	74.53	0.203	0.412
Shear force, kg	3.16	3.23	0.232	0.757
Cook loss, %	26.28	28.16	4.287	0.602
pH	5.45	5.42	0.082	0.444
Objective color				
L*	56.85	56.94	1.26	0.962
a*	8.194	7.938	0.685	0.631
b*	15.609	15.401	0.999	0.786
Subjective evaluations				
Color	1.98	2.12	0.241	0.581
Marbling	1.671	1.671	0.346	1.000
Firmness	2.857	2.429	0.175	0.087
Drip loss, %	3.367	4.259	0.678	0.394

^1^ Each least squares mean represents seven observations.

**Table 4 animals-09-00970-t004:** Carcass cutability of finishing pigs fed control and enzymatically digested food waste.

Item ^1^	Control	Food Waste	SEM	*p*-Value
Half carcass chilled weight, kg	40.68	37.43	1.38	0.054
Lean cutting yield ^2^, %	63.85	65.33	3.80	0.329
Boneless lean cutting yield ^3^, %	48.72	50.25	2.19	0.271
Carcass cutting yield ^4^, %	72.78	74.58	3.94	0.256
Boneless carcass cutting yield ^5^, %	57.65	58.50	2.06	0.203

^1^ Each least squares mean represents seven observations. ^2^ Lean cutting yield, % = [(trimmed ham, kg + trimmed loin, kg + Boston butt, kg + picnic, kg) ÷ half chilled carcass weight, kg] × 100. ^3^ Boneless lean cutting yield, % = [(inside ham, kg + outside ham, kg + knuckle, kg + tenderloin, kg + sirloin, kg + boneless Boston butt, kg + boneless picnic, kg) ÷ half side chilled carcass weight, kg] × 100. ^4^ Carcass cutting yield, % = [(trimmed ham, kg + trimmed loin, kg + Boston butt, kg + picnic, kg + trimmed belly, kg) ÷ half side chilled carcass weight, kg] × 100. ^5^ Boneless carcass cutting yield, % = [(inside ham, kg + outside ham, kg + knuckle, kg + tenderloin, kg + sirloin, kg + boneless Boston butt, kg + boneless picnic, kg + trimmed belly) ÷ half side chilled carcass weight, kg] × 100.

**Table 5 animals-09-00970-t005:** Back-fat chemical composition and fatty acid profile of finishing pigs fed control and enzymatically digested food waste.

Item ^1^	Control	Food Waste	SEM	*p*-Value
Moisture, %	10.41	11.24	1.14	0.54
Crude fat, %	86.00	85.19	1.55	0.60
Saturated fatty acids, % of crude fat				
Myristic (14:0)	1.19	1.216	0.03	0.54
Pentadecylic (15:0)	0.05	0.126	0.01	<0.01
Palmitic (16:0)	22.40	19.482	0.68	<0.01
Margaric (17:0)	0.30	0.555	0.05	<0.01
Stearic (18:0)	11.36	10.372	1.06	0.16
Arachidic (20:0)	0.23	0.182	0.03	<0.01
Behenoic (22:0)	0.040	0.041	0.00	0.62
Total	35.56	31.96	1.66	0.02
Unsaturated fatty acids, % of crude fat				
Myristoleic (14:1)	0.02	0.07	0.01	<0.01
Palmitoleic (16:1)	1.76	2.24	0.17	<0.01
Oleic (18:1)	35.21	39.64	0.68	<0.01
Vaccenic (18:1)	2.36	2.842	0.12	<0.01
Linoleic (18:2)	19.51	15.08	0.64	<0.01
Linolenic (18:3)	1.52	1.21	0.09	<0.01
Gondoic (20:1)	0.68	0.83	0.14	<0.01
Eicosadienoic (20:2)	0.77	0.63	0.04	0.02
Arachidonic (20:4)	0.25	0.26	0.02	0.73
Eicosapentaenoic (EPA, 20:5)	0.01	0.09	0.04	0.02
Docosaherxaenoic (DHA, 22:6)	0.01	0.17	0.06	<0.01
Monounsaturated fatty acids	40.04	45.64	0.83	<0.01
Polyunsaturated fatty acids	22.36	17.84	0.71	<0.01
Total fatty acids, % of crude fat	62.52	63.61	1.34	0.33

^1^ Each least squares mean represents seven observations.

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
