# Peer review of "Enzymatically Digested Food Waste Altered Fecal Microbiota But Not Meat Quality and Carcass Characteristics of Growing-Finishing Pigs"

_animals, 2019, doi:10.3390/ani9110970_

Round 1

Reviewer 1 Report

The present work was well appreciated. However, several blemishes that need revision before this work can be considered for publication.

There are some English grammatical corrections throughout the manuscript. The quality of the English is also a concern. Your manuscript would greatly improve from editing by a native English speaker.

Abstract was not clear to understand, no detailed information about animal and  experimental results.

Background of research is not strong enough.

Line no. 25: i.e., DNA? Is it DHA, right?

Line no. :Experimental design is not explained well in Materials and Methods section. How long the experiment period? Make it out as simple.

Is that animals crossbred? Should mention in text.

Source of liquid diet from enzymatically digested food waste? Explain briefly about liquid feed diet in materials and method section.

Lineno. 90: Pigs were fed dietary treatments in phases 1 and 2, and all pigs were fed with control diet in 91 phase 3. – Why?

            Growth performance – ADG, ADFI, FCR and nutrient digestibility Analysis? – why authors ignore these?

Abbreviations must be defined at first mention and used consistently thereafter in entire manuscript. Revise entire manuscript.

Better to be write Results and Discussion in Separate. Revise the manuscript with strong justification for background. The authors should avoid the lengthy sentences and incomplete sentences.

Author Response

Reviewer 1

The present work was well appreciated. However, several blemishes that need revision before this work can be considered for publication.

There are some English grammatical corrections throughout the manuscript. The quality of the English is also a concern. Your manuscript would greatly improve from editing by a native English speaker.

RESPONSE: The manuscript has been thoroughly revised again by all co-authors, which included couple native English speakers. We hope the current version has great improvement compared with the original one.

Abstract was not clear to understand, no detailed information about animal and experimental results.

RESPONSE: Abstract has included research aim, brief materials and methods, main findings in meat quality and fecal microbiome, and a conclusion. Due to the word limitation, it is difficult to include more details about materials and methods. We truly appreciate your understanding.

Background of research is not strong enough.

RESPONSE: More information about food waste processing procedures was added to the revised manuscript (L57-61). Hope the revised manuscript provides more clear research background and objectives compared with the original version.

Line no. 25: i.e., DNA? Is it DHA, right?

RESPONSE: Changed as suggested (L26).

Line no. :Experimental design is not explained well in Materials and Methods section. How long the experiment period? Make it out as simple.

RESPONSE: The experimental design was clearly described in our previously published article (Jinno et al., 2018). We have modified the Materials and Methods as suggested. We believe all required information was provided in the revised manuscript.

Is that animals crossbred? Should mention in text.

RESPONSE: Changed as suggested (L87).

Source of liquid diet from enzymatically digested food waste? Explain briefly about liquid feed diet in materials and method section.

RESPONSE: More information was added to the revised manuscript as suggested (L57-61).

Lineno. 90: Pigs were fed dietary treatments in phases 1 and 2, and all pigs were fed with control diet in 91 phase 3. – Why?

RESPONSE: The enzymatically digested food waste contained very high concentration of fat, which would be a good product for sows and pigs less than 80 kg. We switched food waste diet to control on phase 3 (late-finishing pigs) to avoid soft fat in pork. The related information was also added to the revised manuscript. The growth performance results that were published in Jinno et al. (2018) confirmed the correctness of experimental design.

            Growth performance – ADG, ADFI, FCR and nutrient digestibility Analysis? – why authors ignore these?

RESPONSE: The data for growth performance have been published in Jinno et al. (2018). We also clarified it in the revised manuscript.

Abbreviations must be defined at first mention and used consistently thereafter in entire manuscript. Revise entire manuscript.

RESPONSE: Double checked throughout the manuscript. Redundant abbreviations were removed from the manuscript. Thank you very much for the comment.

Better to be write Results and Discussion in Separate. Revise the manuscript with strong justification for background. The authors should avoid the lengthy sentences and incomplete sentences.

RESPONSE: More information was added to the introduction section. The manuscript was thoroughly revised as suggested and to correct any grammar mistake. We truly appreciate your suggestion, but we prefer to keep Results and Discussion in one section to avoid repeating results’ description in the discussion section.

Reviewer 2 Report

In current study growing-finishing pigs were fed with enzymatically digested food waste and its effect on meat quality, fatty acid profile of back-fat and fecal microbiota were analyzed. It was determined that enzymatically digested waste diet did not affect the meat quality, enhanced the beneficial fatty acid contents and altered fecal microbiota. Although, it is a well-designed study with huge importance, it still needs to answer few important questions as following before publishing

Diet increased the abundances of Lachnospiraceae and Ruminococcaceae but decreased the relative abundances of Streptococcaceae and Clostridiaceae. Give the data of diversity in terms of both of the numerical and structural diversity. Line 74: What is approval number of Institutional Animal Care and Use Committee at the University of California, Davis (UC Davis)? What is overall growth performance of pigs fed on different diets Was toxicity profile of the food waste feed was determined, if yes include that data. Safety of the food waste food needs to be established What is the effect of food waste feed on organo-leptic and acceptability by consumer limitations of the study?

Author Response

Reviewer 2

In current study growing-finishing pigs were fed with enzymatically digested food waste and its effect on meat quality, fatty acid profile of back-fat and fecal microbiota were analyzed. It was determined that enzymatically digested waste diet did not affect the meat quality, enhanced the beneficial fatty acid contents and altered fecal microbiota. Although, it is a well-designed study with huge importance, it still needs to answer few important questions as following before publishing.

Diet increased the abundances of Lachnospiraceae and Ruminococcaceae but decreased the relative abundances of Streptococcaceae and Clostridiaceae. Give the data of diversity in terms of both of the numerical and structural diversity.

RESPONSE: First of all, we apologize if we misunderstood the question. The data generated from 16S rRNA sequencing were relative data, therefore, the abundance of individual bacterial phylum, family, or species was relative value. We have added the values (relative abundance as percentage) in the revised manuscript (361-368). Please let us know if this made sense for you. Thank you very much.

Line 74: What is approval number of Institutional Animal Care and Use Committee at the University of California, Davis (UC Davis)?

RESPONSE: The IACUC # was added to the revised manuscript (L83).

What is overall growth performance of pigs fed on different diets?

RESPONSE: The growth performance data have been published in Jinno et al. (2018). We have added the brief summarization of performance data in the discussion section of the revised manuscript (L215-219).

Was toxicity profile of the food waste feed was determined, if yes include that data. Safety of the food waste food needs to be established? What is the effect of food waste feed on organo-leptic and acceptability by consumer limitations of the study? 

RESPONSE: We did not measure the toxicity profile of the food waste feed. However, the company who produced and provided the food waste product for the animal trial is doing routine test to verify the free of foodborne pathogens in the final product. The related reference (Pandey et al., 2016) was also added to the revised manuscript.

The organoleptic tests were not conducted in the current experiment since this was a pilot study to test the potential of using food waste as growing-finishing pigs. We are still working on the procedures to dry the enzymatically food waste, which will increase the product shelf-life and reduce the transportation cost. A large scale animal trial will be planned to test the optimal inclusion rate of this food waste product in swine diet at different physiological stages.

Reviewer 3 Report

The avoidance of food waste has environmental implications and using it in animal feed after enzymatic processing could be a good idea, especially for small farms. However because of ASF it is forbidden in many countries. Food waste are difficult feed mixture ingredient because of unstable nutritive value.

Presented paper is prepared correctly, but I have some objectives to methods and as a consequence of it to some discussion aspects and conclusions.

Simple summary and Abstract look interesting and encouraging. Introduction like above, but ..... After reading this part I found " Enzymatic digestion  turns food waste into feed for growing pigs" (Jinno et al. 2018), read it and I was surprised. I back to the revised paper.
I have serious objectives to the first part of Material and Methods: 2.1. Animals, husbandry, experimental design, and dietary treatments. The experimental factor isn't clear for me, generally it is feed but authors have more experimental factors that they point:

1) different feeding: dry (control group -C) and wet (experimental group -E)

2) different feeding programme: 3 phase in C and 2 phase in E

3) different nutrition value of feed in both groups

all these factors influencing on microbiom and slaughtering performance of pigs and it should be taken into account in the discussion.

So however the rest of presented methods is correct, mentioned reservations as a lack of clarity regarding the impact of clearly identified experimental factor may result in erroneous interpretation.
Maybe my objectives are unfounded and authors can answer me:

- is described experiment the new one or just it is a continuation of it, described in Animal Feed Science and Technology (2018)?, we have exactly the same feed components and nutritive value,

- how many times food waste nutrition value was analyzed?, I suppose that it can't be stable through all the experimental time, I wonder how big (if were) these changes were.

            I am confused because mentioned earlier article is accepted with the same methodological mistake - wrong division into experimental groups or wrong description of the experimental factor.

I don't want to write that the work done by authors is wrong (presenting results are interersting), I'll wait for answer. Perhaps it will be better just to describe only results obtained in experimental group and to compare them with described in literature or to change a little the aim of the study and explain why in one study you compare the final effect of so different feeding technology (form, phase number, nutritive value of feed mixtures). Authors have very good conditions to estimate these different factors in one study: four pens and four group in one replication.

Author Response

Reviewer 3

The avoidance of food waste has environmental implications and using it in animal feed after enzymatic processing could be a good idea, especially for small farms. However, because of ASF it is forbidden in many countries. Food waste are difficult feed mixture ingredient because of unstable nutritive value.

RESPONSE: In Jinno et al. (2018), the chemical composition of this food waste product was analyzed with samples collected from 11 batches. The results confirm the current enzymatic digestion method could produce consistent food waste product. This was done before we planned the animal trial in growing-finishing pigs.

Presented paper is prepared correctly, but I have some objectives to methods and as a consequence of it to some discussion aspects and conclusions.

Simple summary and Abstract look interesting and encouraging. Introduction like above, but ..... After reading this part, I found " Enzymatic digestion  turns food waste into feed for growing pigs" (Jinno et al. 2018), read it and I was surprised. I back to the revised paper. 
I have serious objectives to the first part of Material and Methods: 2.1. Animals, husbandry, experimental design, and dietary treatments. The experimental factor isn't clear for me, generally it is feed but authors have more experimental factors that they point:

1) different feeding: dry (control group -C) and wet (experimental group -E)

2) different feeding programme: 3 phase in C and 2 phase in E

3) different nutrition value of feed in both groups

all these factors influencing on microbiome and slaughtering performance of pigs and it should be taken into account in the discussion.

RESPONSE: we really appreciate your comments on the experimental design. We have modified the 2.1. of Materials and Methods. Hope the experimental design was clear in the revised manuscript.

The experimental design was a simple Randomized Complete Block Design with 2 dietary treatments: control vs. food waste. The experiment was conducted 79 days with 3-feeding phase program as we described in 2.1. Pigs in both treatments all had 3 feeding phases. In phases 1 and 2, pigs received either control or food waste. In phase 3, all pigs received finisher-2 control diet. The reason was to avoid soft fat in pork since the enzymatically digested food waste contained very high concentration of fat.

In regard to the question for nutrition values, we formulated the control diets based on NRC 2012. All diets met the requirement of growing-finishing pigs. Based on the chemical composition analysis, food waste contained more nutrients and energy than control diets. Therefore, when we initially designed the experiment, we expected the pigs fed with food waste grew faster than pigs in control although the performance results were not same as we expected (Jinno et al., 2018).

We completely agree that the form of the diet could affect gut microbiome, however there was limited research exploring this hypothesis so far. We have added the related information to the revised manuscript.

So however the rest of presented methods is correct, mentioned reservations as a lack of clarity regarding the impact of clearly identified experimental factor may result in erroneous interpretation.

Maybe my objectives are unfounded and authors can answer me:

- is described experiment the new one or just it is a continuation of it, described in Animal Feed Science and Technology (2018)? we have exactly the same feed components and nutritive value,

RESPONSE: The animal trial described in the current experiment was the same one as Jinno et al. (2018) with different research focuses. We are sorry for the confusion. We have clarified this information in the revised manuscript.

- how many times food waste nutrition value was analyzed?, I suppose that it can't be stable through all the experimental time, I wonder how big (if were) these changes were.

RESPONSE: In Jinno et al. (2018), the chemical compositions of this food waste product were analyzed with samples collected from 11 batches. The results confirmed the current enzymatic digestion method could produce consistent food waste product. This analytical work was done before we started to plan the animal trial with growing-finishing pigs.

During the animal trial, food waste was freshly produced every week and subsampled. The food waste subsamples were pooled within feeding phase and were analyzed again for proximal analysis and fatty acid profile. The nutritional values were very consistent. We have added the related information to the revised manuscript.

            I am confused because mentioned earlier article is accepted with the same methodological mistake - wrong division into experimental groups or wrong description of the experimental factor.

RESPONSE: The animal trial was a small scale and pilot study to evaluate if the enzymatically digested food waste can be utilized by growing-finishing pigs. We chose to use a very simple experimental design and we believe the design was appropriate to test the objectives of the research.

I don't want to write that the work done by authors is wrong (presenting results are interesting), I'll wait for answer. Perhaps it will be better just to describe only results obtained in experimental group and to compare them with described in literature or to change a little the aim of the study and explain why in one study you compare the final effect of so different feeding technology (form, phase number, nutritive value of feed mixtures). Authors have very good conditions to estimate these different factors in one study: four pens and four group in one replication.

RESPONSE: please see the responses above. We hope the revised manuscript provide clear description about the animal trial and our experimental design.

Round 2

Reviewer 1 Report

Appreciate the authors for their prompt action to revise the manuscript.

All comments were answered, authors satisfy with their justification in revised manuscript.

Author Response

We truly appreciate the valuable comments you have provided, which were very helpful to improve our original manuscript. 

Thank you very much for your time and effort to review this manuscript.

Reviewer 3 Report

I am satisfied with the response to my comments. Generally i like this paper and recommend to print.

Check:

line 37/38 Meat quality .... is this sentence correct?, shuoldn't be added "in" 

line 210 shouldn't be space between pairwise and Adonise?

line 409 shouldn't be "change"?

And one suggestion for future: I wrote earlier that according to feeding you had to experimental factors: dry and feed (and of course different nutritive value) and the second one in control group you had 3 diets and in experimental 2. Change it in next studies it will better. 

Good luck!, this work is good.

Author Response

I am satisfied with the response to my comments. Generally I like this paper and recommend to print.

Check:

line 37/38 Meat quality .... is this sentence correct?, shuoldn't be added "in" 

RESPONSE: Corrected as suggested (L38).

line 210 shouldn't be space between pairwise and Adonise?

RESPONSE: The content was not changed. The pairwiseAdonis is the correct name for a R package.

line 409 shouldn't be "change"?

RESPONSE: Corrected as suggested (L409).

And one suggestion for future: I wrote earlier that according to feeding you had to experimental factors: dry and feed (and of course different nutritive value) and the second one in control group you had 3 diets and in experimental 2. Change it in next studies it will better. 

Good luck!, this work is good.

RESPONSE: We truly appreciate all the valuable comments you have provided to us. Those comments were very helpful to improve our original manuscript writing and to help us design new experiment in the future research. Thank you very much for your time and effort to revise this manuscript.